# Deconvolution of High Dimensional Mixtures via Boosting, with Application to Diffusion-Weighted MRI of Human Brain

**Charles Y. Zheng**
Department of Statistics
Stanford University
Stanford, CA 94305
snarles@stanford.edu

**Franco Pestilli**
Department of Psychological and Brain Sciences
Indiana University, Bloomington, IN 47405
franpest@indiana.edu

**Ariel Rokem**
Department of Psychology
Stanford University
Stanford, CA 94305
arokem@stanford.edu

## Abstract

Diffusion-weighted magnetic resonance imaging (DWI) and fiber tractography are the only methods to measure the structure of the white matter in the living human brain. The diffusion signal has been modelled as the combined contribution from many individual fascicles of nerve fibers passing through each location in the white matter. Typically, this is done via *basis pursuit*, but estimation of the exact directions is limited due to discretization [1, 2]. The difficulties inherent in modeling DWI data are shared by many other problems involving fitting non-parametric mixture models. Ekanadaham et al. [3] proposed an approach, *continuous basis pursuit*, to overcome discretization error in the 1-dimensional case (e.g., spike-sorting). Here, we propose a more general algorithm that fits mixture models of any dimensionality without discretization. Our algorithm uses the principles of L2-boost [4], together with refitting of the weights and pruning of the parameters. The addition of these steps to L2-boost both accelerates the algorithm and assures its accuracy. We refer to the resulting algorithm as *elastic basis pursuit*, or EBP, since it expands and contracts the active set of kernels as needed. We show that in contrast to existing approaches to fitting mixtures, our boosting framework (1) enables the selection of the optimal bias-variance tradeoff along the solution path, and (2) scales with high-dimensional problems. In simulations of DWI, we find that EBP yields better parameter estimates than a non-negative least squares (NNLS) approach, or the standard model used in DWI, the tensor model, which serves as the basis for diffusion tensor imaging (DTI) [5]. We demonstrate the utility of the method in DWI data acquired in parts of the brain containing crossings of multiple fascicles of nerve fibers.

# 1 Introduction

In many applications, one obtains measurements $(x_i, y_i)$ for which the response $y$ is related to $x$ via some mixture of known kernel functions $f_\theta(x)$, and the goal is to recover the mixture parameters $\theta_k$ and their associated weights:

$$y_i = \sum_{k=1}^{K} w_k f_{\theta_k}(x) + \epsilon_i \tag{1}$$

where $f_\theta(x)$ is a known kernel function parameterized by $\theta$, and $\boldsymbol{\theta} = (\theta_1, \ldots, \theta_K)$ are model parameters to be estimated, $w = (w_1, \ldots, w_K)$ are unknown nonnegative weights to be estimated, and $\epsilon_i$ is additive noise. The number of components $K$ is also unknown, hence, this is a *nonparametric model*. One example of a domain in which mixture models are useful is the analysis of data from diffusion-weighted magnetic resonance imaging (DWI). This biomedical imaging technique is sensitive to the direction of water diffusion within millimeter-scale voxels in the human brain *in vivo*. Water molecules freely diffuse along the length of nerve cell axons, but is restricted by cell membranes and myelin along directions orthogonal to the axon's trajectory. Thus, DWI provides information about the microstructural properties of brain tissue in different locations, about the trajectories of organized bundles of axons, or fascicles within each voxel, and about the connectivity structure of the brain. Mixture models are employed in DWI to deconvolve the signal within each voxel with a kernel function, $f_\theta$, assumed to represent the signal from every individual fascicle [1, 2] (Figure 1B), and $w_i$ provide an estimate of the fiber orientation distribution function (fODF) in each voxel, the direction and volume fraction of different fascicles in each voxel. In other applications of mixture modeling these parameters represent other physical quantities. For example, in chemometrics, $\theta$ represents a chemical compound and $f_\theta$ its spectra. In this paper, we focus on the application of mixture models to the data from DWI experiments and simulations of these experiments.

## 1.1 Model fitting - existing approaches

Hereafter, we restrict our attention to the use of squared-error loss; resulting in penalized least-squares problem

$$\text{minimize }_{\hat{K}, \hat{w}, \hat{\boldsymbol{\theta}}} \left\| y_i - \sum_{k=1}^{\hat{K}} \hat{w}_k f_{\hat{\theta}_k}(x_i) \right\|^2 + \lambda P_{\boldsymbol{\theta}}(w) \tag{2}$$

Minimization problems of the form (2) can be found in the signal deconvolution literature and elsewhere: some examples include super-resolution in imaging [6], entropy estimation for discrete distributions [7], X-ray diffraction [8], and neural spike sorting [3]. Here, $P_{\boldsymbol{\theta}}(w)$ is a *convex* penalty function of $(\boldsymbol{\theta}, w)$. Examples of such penalty functions given in Section 2.1; a formal definition of convexity in the nonparametric setting can be found in the supplementary material, but will not be required for the results in the paper. Technically speaking, the objective function (2) is convex in $(w, \boldsymbol{\theta})$, but since its domain is of infinite dimensionality, for all practical purposes (2) is a nonconvex optimization problem. One can consider fixing the number of components in advance, and using a descent method (with random restarts) to find the best model of that size. Alternatively, one could use a stochastic search method, such as simulated annealing or MCMC [9], to estimate the size of the model and the model parameters simultaneously. However, as one begins to consider fitting models with increasing number of components $\hat{K}$ and of high dimensionality, it becomes increasingly difficult to apply these approaches [3]. Hence a common approach to obtaining an approximate solution to (2) is to limit the search to a discrete grid of candidate parameters $\boldsymbol{\theta} = \theta_1, \ldots, \theta_p$. The estimated weights and parameters are then obtained by solving an optimization problem of the form

$$\hat{\beta} = \text{argmin}_{\beta > 0} ||y - \vec{F}\beta||^2 + \lambda P_{\boldsymbol{\theta}}(\beta)$$

where $\vec{F}$ has the $j$th column $\vec{f}_{\theta_j}$, where $\vec{f}_\theta$ is defined by $(\vec{f}_\theta)_i = f_\theta(x_i)$. Examples applications of this non-negative least-squares-based approach (NNLS) include [10] and [1, 2, 7]. In contrast to descent based methods, which get trapped in local minima, NNLS is guaranteed to converge to a solution which is within $\epsilon$ of the global optimum, where $\epsilon$ depends on the scale of discretization. In

some cases, NNLS will predict the signal accurately (with small error), but the parameters resulting will still be erroneous. Figure 1 illustrates the worst-case scenario where discretization is misaligned relative to the true parameters/kernels that generated the signal.

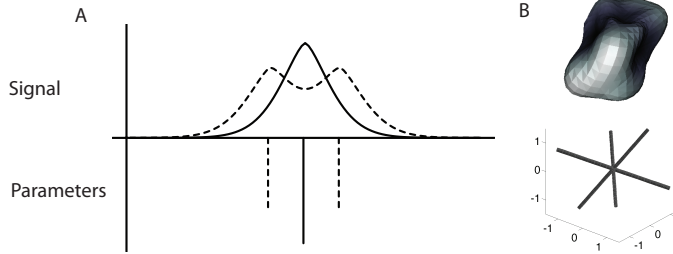

Figure 1: The signal deconvolution problem. Fitting a mixture model with a NNLS algorithm is prone to errors due to discretization. For example, in 1D (A), if the true signal (top; dashed line) arises from a mixture of signals from a bell-shaped kernel functions (bottom; dashed line), but only a single kernel function between them is present in the basis set (bottom; solid line), this may result in inaccurate signal predictions (top; solid line), due to erroneous estimates of the parameters $w_i$. This problem arises in deconvolving multi-dimensional signals, such as the 3D DWI signal (B), as well. Here, the DWI signal in an individual voxel is presented as a 3D surface (top). This surface results from a mixture of signals arising from the fascicles presented on the bottom passing through this single (simulated) voxel. Due to the signal generation process, the kernel of the diffusion signal from each one of the fascicles has a minimum at its center, resulting in 'dimples' in the diffusion signal in the direction of the peaks in the fascicle orientation distribution function.

In an effort to improve the discretization error of NNLS, Ekanadham et al [3] introduced continuous basis pursuit (CBP). CBP is an extension of nonnegative least squares in which the points on the discretization grid $\theta_1, \ldots, \theta_p$ can be continuously moved within a small distance; in this way, one can reach any point in the parameter space. But instead of computing the actual kernel functions for the perturbed parameters, CBP uses linear approximations, e.g. obtained by Taylor expansions. Depending on the type of approximation employed, CBP may incur large error. The developers of CBP suggest solutions for this problem in the one-dimensional case, but these solutions cannot be used for many applications of mixture models (e.g DWI). The computational cost of both NNLS and CBP scales exponentially in the dimensionality of the parameter space. In contrast, using stochastic search methods or descent methods to find the global minimum will generally incur a computational cost scaling which is exponential in the sample size times the parameter space dimensions. Thus, when fitting high-dimensional mixture models, practitioners are forced to choose between the discretization errors inherent to NNLS, or the computational difficulties in the descent methods. We will show that our boosting approach to mixture models combines the best of both worlds: while it does not suffer from discretization error, it features computational tractability comparable to NNLS and CBP. We note that for the specific problem of super-resolution, Càndes derived a deconvolution algorithm which finds the global minimum of (2) without discretization error and proved that the algorithm can recover the true parameters under a minimal separation condition on the parameters [6]. However, we are unaware of an extension of this approach to more general applications of mixture models.

### 1.2 Boosting

The model (1) appears in an entirely separate context, as the model for learning a regression function as an ensemble of weak learners $f_\theta$, or boosting [4]. However, the problem of fitting a mixture model and the problem of fitting an ensemble of weak learners have several important differences. In the case of learning an ensemble, the family $\{f_\theta\}$ can be freely chosen from a universe of possible weak learners, and the only concern is minimizing the prediction risk on a new observation. In contrast, in the case of fitting a mixture model, the family $\{f_\theta\}$ is specified by the application. As a result, boosting algorithms, which were derived under the assumption that $\{f_\theta\}$ is a suitably flexible class of weak learners, generally perform poorly in the signal deconvolution setting, where the family $\{f_\theta\}$ is inflexible. In the context of regression, $L_2$boost, proposed by Buhlmann et al [4] produces a

path of ensemble models which progressively minimize the sum of squares of the residual. $L_2$boost fits a series of models of increasing complexity. The first model consists of the single weak learner $\vec{f_\theta}$ which best fits $y$. The second model is formed by finding the weak learner with the greatest correlation to the residual of the first model, and adding the new weak learner to the model, without changing any of the previously fitted weights. In this way the size of the model grows with the number of iterations: each new learner is fully fit to the residual and added to the model. But because the previous weights are never adjusted, $L_2$Boost fails to converge to the global minimum of (2) in the mixture model setting, producing suboptimal solutions. In the following section, we modify $L_2$Boost for fitting mixture models. We refer to the resulting algorithm as *elastic basis pursuit*.

## 2 Elastic Basis Pursuit

Our proposed procedure for fitting mixture models consists of two stages. In the first stage, we transform a $L_1$ penalized problem to an equivalent *non regularized* least squares problem. In the second stage, we employ a modified version of $L_2$Boost, *elastic basis pursuit*, to solve the transformed problem. We will present the two stages of the procedure, then discuss our fast convergence results.

### 2.1 Regularization

For most mixture problems it is beneficial to apply a $L_1$-norm based penalty, by using a modified input $\tilde{y}$ and kernel function family $\tilde{f_\theta}$, so that

$$\text{argmin}_{K,w,\boldsymbol{\theta}} \left\| y - \sum_{i=1}^{K} \vec{f_\theta} \right\|^2 + \lambda P_{\boldsymbol{\theta}}(w) = \text{argmin}_{K,w,\boldsymbol{\theta}} \left\| \tilde{y} - \sum_{i=1}^{K} \tilde{f_\theta} \right\|^2 \qquad (3)$$

We will use our modified $L_2$Boost algorithm to produce a path of solutions for objective function on the left side, which results in a solution path for the penalized objective function (2).

For example, it is possible to embed the penalty $P_{\boldsymbol{\theta}}(w) = ||w||_1^2$ in the optimization problem (2). One can show that solutions obtained by using the penalty function $P_{\boldsymbol{\theta}}(w) = ||w||_1^2$ have a one-to-one correspondence with solutions of obtained using the usual $L_1$ penalty $||w||_1$. The penalty $||w||_1^2$ is implemented by using the transformed input: $\tilde{y} = \begin{pmatrix} y \\ 0 \end{pmatrix}$ and using modified kernel vectors $\tilde{f_\theta} = \begin{pmatrix} \vec{f_\theta} \\ \sqrt{\lambda} \end{pmatrix}$. Other kinds of regularization are also possible, and are presented in the *supplemental material*.

### 2.2 From $L_2$Boost to Elastic Basis Pursuit

Motivated by the connection between boosting and mixture modelling, we consider application of $L_2$Boost to solve the transformed problem (the left side of(3)). Again, we reiterate the *nonparametric* nature of the model space; by minimizing (3), we seek to find the model with *any* number of components which minimizes the residual sum of squares. In fact, given appropriate regularization, this results in a well-posed problem. In each iteration of our algorithm a subset of the parameters, $\boldsymbol{\theta}$ are considered for adjustment. Following Lawson and Hanson [11], we refer to these as the *active set*. As stated before, $L_2$Boost can only grow the active set at each iteration, converging to inaccurate models. Our solution to this problem is to modify $L_2$Boost so that it grows *and* contracts the active set as needed; hence we refer to this modification of the $L_2$Boost algorithm as *elastic basis pursuit*. The key ingredient for any boosting algorithm is an oracle for fitting a weak learner: that is, a function $\tau$ which takes a residual as input and returns the parameter $\theta$ corresponding to the kernel $\tilde{f_\theta}$ most correlated with the residual. EBP takes as inputs the oracle $\tau$, the input vector $\tilde{y}$, the function $\tilde{f_\theta}$, and produces a path of solutions which progressively minimize (3). To initialize the algorithm, we use NNLS to find an initial estimate of $(w, \boldsymbol{\theta})$. In the $k$th iteration of the boosting algorithm, let $\tilde{r}^{(k-1)}$ be residual from the previous iteration (or the NNLS fit, if $k = 1$). The algorithm proceeds as follows

1. Call the oracle to find $\theta_{new} = \tau(\tilde{r}^{(k-1)})$, and add $\theta_{new}$ to the active set $\boldsymbol{\theta}$.

2. Refit the weights $w$, using NNLS, to solve:

$$\text{minimize}_{w>0}||\tilde{y} - \tilde{F}w||^2$$

   where $\tilde{F}$ is the matrix formed from the regressors in the active set, $\tilde{f}_\theta$ for $\theta \in \boldsymbol{\theta}$. This yields the residual $\tilde{r}^{(k)} = \tilde{y} - \tilde{F}w$.

3. Prune the active set $\boldsymbol{\theta}$ by removing any parameter $\theta$ whose weight is zero, and update the weight vector $w$ in the same way. This ensures that the active set $\boldsymbol{\theta}$ remains sparse in each iteration. Let $(w^{(k)}, \boldsymbol{\theta}^{(k)})$ denote the values of $(w, \boldsymbol{\theta})$ at the end of this step of the iteration.

4. Stopping may be assessed by computing an estimated prediction error at each iteration, via an independent validation set, and stopping the algorithm early when the prediction error begins to climb (indicating overfitting).

Psuedocode and Matlab code implementing this algorithm can be found in the supplement.

In the boosting context, the property of refitting the ensemble weights in every iteration is known as the *totally corrective* property; LPBoost [12] is a well-known example of a totally corrective boosting algorithm. While we derived EBP as a totally corrective variant of $L_2$Boost, one could also view EBP as a generalization of the classical Lawson-Hanson (LH) algorithm [11] for solving nonnegative least-squares problems. Given mild regularity conditions and appropriate regularization, Elastic Basis Pursuit can be shown to deterministically converge to the global optimum: we can bound the objective function gap in the $m$th iteration by $C/\sqrt{m}$, where $C$ is an explicit constant (see 2.3). To our knowledge, fixed iteration guarantees are unavailable for all other methods of comparable generality for fitting a mixture with an unknown number of components.

## 2.3 Convergence Results

*(Detailed proofs can be found in the supplementary material.)*

For our convergence results to hold, we require an oracle function $\tau : \mathbb{R}^{\tilde{n}} \rightarrow \Theta$ which satisfies

$$\left\langle \tilde{r}, \frac{\tilde{f}_{\tau(\tilde{r})}}{||\tilde{f}_{\tau(\tilde{r})}||} \right\rangle \geq \alpha\rho(\tilde{r}), \text{ where } \rho(\tilde{r}) = \sup_{\theta \in \Theta} \left\langle \tilde{r}, \frac{\tilde{f}_\theta}{||\tilde{f}_\theta||} \right\rangle \tag{4}$$

for some fixed $0 < \alpha <= 1$. Our proofs can also be modified to apply given a stochastic oracle that satisfies (4) with fixed probability $p > 0$ for every input $\tilde{r}$. Recall that $\tilde{y}$ denotes the transformed input, $\tilde{f}_\theta$ the transformed kernel and $\tilde{n}$ the dimensionality of $\tilde{y}$. We assume that the parameter space $\Theta$ is compact and that $\tilde{f}_\theta$, the transformed kernel function, is continuous in $\theta$. Furthermore, we assume that either $L_1$ regularization is imposed, *or* the kernels satisfy a positivity condition, i.e. $\inf_{\theta \in \Theta} f_\theta(x_i) \geq 0$ for $i = 1, \ldots, n$. Proposition 1 states that these conditions imply the existence of a maximally saturated model $(w^*, \boldsymbol{\theta}^*)$ of size $K^* \leq \tilde{n}$ with residual $\tilde{r}^*$.

The existence of such a saturated model, in conjunction with existence of the oracle $\tau$, enables us to state fixed-iteration guarantees on the precision of EBP, which implies asymptotic convergence to the global optimum. To do so, we first define the quantity $\rho^{(m)} = \rho(\tilde{r}^{(m)})$, see (4) above. Proposition 2 uses the fact that the residuals $\tilde{r}^{(m)}$ are orthogonal to $\tilde{F}^{(m)}$, thanks to the NNLS fitting procedure in step 2. This allows us to bound the objective function gap in terms of $\rho^{(m)}$. Proposition 3 uses properties of the oracle $\tau$ to lower bound the progress per iteration in terms of $\rho^{(m)}$.

**Proposition 2** *Assume the conditions of Proposition 1. Take saturated model $w^*, \boldsymbol{\theta}^*$. Then defining*

$$B^* = 2 \sum_{i=1}^{K^*} w_i^* ||\tilde{f}_{\theta_i^*}|| \tag{5}$$

*the $m$th residual of the EBP algorithm $\tilde{r}^{(m)}$ can be bounded in size by*

$$||\tilde{r}^{(m)}||^2 \leq ||\tilde{r}^*||^2 + B^*\rho^{(m)}$$

In particular, whenever $\rho$ converges to 0, the algorithm converges to the global minimum.

**Proposition 3** *Assume the conditions of Proposition 1. Then*

$$||\tilde{r}^{(m)}||^2 - ||\tilde{r}^{(m+1)}||^2 \geq (\alpha\rho^{(m)})^2$$

*for $\alpha$ defined above in* (4)*. This implies that the sequence $||\tilde{r}^{(0)}||^2, \ldots$ is decreasing.*

Combining Propositions 2 and 3 yields our main result for the non-asymptotic convergence rate.

**Proposition 4** *Assume the conditions of Proposition 1. Then for all $m > 0$,*

$$||\tilde{r}^{(m)}||^2 - ||\tilde{r}^*||^2 \leq \frac{B_{min}\sqrt{||\tilde{r}^{(0)}||^2 - ||\tilde{r}^*||^2||}}{\alpha}\frac{1}{\sqrt{m}}$$

*where*

$$B_{min} = \inf_{w^*, \boldsymbol{\theta}^*} B^*$$

*for $B^*$ defined in* (5)

Hence we have characterized the non-asymptotic convergence of EBP at rate $\frac{1}{\sqrt{m}}$ with an explicit constant, which in turn implies asymptotic convergence to the global minimum.

## 3  DWI Results and Discussion

To demonstrate the utility of EBP in a real-world application, we used this algorithm to fit mixture models of DWI. Different approaches are taken to modeling the DWI signal. The classical Diffusion Tensor Imaging (DTI) model [5], which is widely used in applications of DWI to neuroscience questions, is not a mixture model. Instead, it assumes that diffusion in the voxel is well approximated by a 3-dimensional Gaussian distribution. This distribution can be parameterized as a rank-2 tensor, which is expressed as a 3 by 3 matrix. Because the DWI measurement has antipodal symmetry, the tensor matrix is symmetric, and only 6 independent parameters need to be estimated to specify it. DTI is accurate in many places in the white matter, but its accuracy is lower in locations in which there are multiple crossing fascicles of nerve fibers. In addition, it should not be used to generate estimates of connectivity through these locations. This is because the peak of the fiber orientation distribution function (fODF) estimated in this location using DTI is not oriented towards the direction of any of the crossing fibers. Instead, it is usually oriented towards an intermediate direction (Figure 4B). To address these challenges, mixture models have been developed, that fit the signal as a combination of contributions from fascicles crossing through these locations. These models are more accurate in fitting the signal. Moreover, their estimate of the fODF is useful for tracking the fascicles through the white matter for estimates of connectivity. However, these estimation techniques either use different variants of NNLS, with a discrete set of candidate directions [2], or with a spherical harmonic basis set [1], or use stochastic algorithms [9]. To overcome the problems inherent in these techniques, we demonstrate here the benefits of using EBP to the estimation of a mixture models of fascicles in DWI. We start by demonstrating the utility of EBP in a simulation of a known configuration of crossing fascicles. Then, we demonstrate the performance of the algorithm in DWI data.

The DWI measurements for a single voxel in the brain are $y_1, \ldots, y_n$ for directions $x_1, \ldots, x_n$ on the three dimensional unit sphere, given by

$$y_i = \sum_{k=1}^{K} w_k f_{D_k}(x_i) + \epsilon_i, \text{ where } f_D(x) = \exp[-bx^T Dx], \tag{6}$$

The kernel functions $f_D(x)$ each describe the effect of a single fascicle traversing the measurement voxel on the diffusion signal, well described by the Stejskal-Tanner equation [13]. Because of the non-negative nature of the MRI signal, $\epsilon_i > 0$ is generated from a Rician distribution [14]. where $b$ is a scalar quantity determined by the experimenter, and related to the parameters of the measurement (the magnitude of diffusion sensitization applied in the MRI instrument). $D$ is a positive definite quadratic form, which is specified by the direction along which the fascicle represented by $f_D$ traverses the voxel and by additional parameters $\lambda_1$ and $\lambda_2$, corresponding to the axial and radial

diffusivity of the fascicle represented by $f_D$. The oracle function $\tau$ is implemented by Newton-Raphson with random restarts. In each iteration of the algorithm, the parameters of $D$ (direction and diffusivity) are found using the oracle function, $\tau(\tilde{r})$, using gradient descent on $\tilde{r}$, the current residuals. In each iteration, the set of $f_D$ is shrunk or expanded to best match the signal.

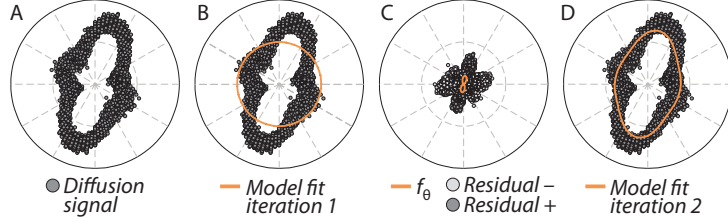

Figure 2: To demonstrate the steps of EBP, we examine data from 100 iterations of the DWI simulation. (A) A cross-section through the data. (B) In the first iteration, the algorithm finds the best single kernel to represent the data (solid line: average kernel). (C) The residuals from this fit (positive in dark gray, negative in light gray) are fed to the next step of the algorithm, which then finds a second kernel (solid line: average kernel). (D) The signal is fit using both of these kernels (which are the *active set* at this point). The combination of these two kernels fits the data better than any of them separately, and they are both kept (solid line: average fit), but redundant kernels can also be discarded at this point (D).

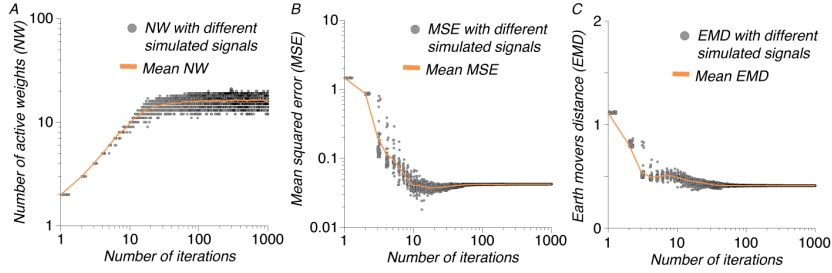

Figure 3: The progress of EBP. In each plot, the abscissa denotes the number of iterations in the algorithm (in log scale). (A) The number of kernel functions in the active set grows as the algorithm progresses, and then plateaus. (B) Meanwhile, the mean square error (MSE) decreases to a minimum and then stabilizes. The algorithm would normally be terminated at this minimum. (C) This point also coincides with a minimum in the optimal bias-variance trade-off, as evidenced by the decrease in EMD towards this point.

In a simulation with a complex configuration of fascicles, we demonstrate that accurate recovery of the true fODF can be achieved. In our simulation model, we take $b = 1000s/mm^2$, and generate $v_1, v_2, v_3$ as uniformly distributed vectors on the unit sphere and weights $w_1, w_2, w_3$ as i.i.d. uniformly distributed on the interval $[0, 1]$. Each $v_i$ is associated with a $\lambda_{1,i}$ between 0.5 and 2, and setting $\lambda_{2,i}$ to 0. We consider the signal in 150 measurement vectors distributed on the unit sphere according to an electrostatic repulsion algorithm. We partition the vectors into a training partition and a test partition to minimize the maximum angular separation in each partition. $\sigma^2 = 0.005$ we generate a signal

We use cross-validation on the training set to fit NNLS with varying L1 regularization parameter $c$, using the regularization penalty function: $\lambda P(w) = \lambda(c - ||w||_1)^2$. We choose this form of penalty function because we interpret the weights $w$ as comprising partial volumes in the voxel; hence $c$ represents the total volume of the voxel weighted by the isotropic component of the diffusion. We fix the regularization penalty parameter $\lambda = 1$. The estimated fODFs and predicted signals are obtained by three algorithms: DTI, NNLS, and EBP. Each algorithm is applied to the training set (75 directions), and error is estimated, relative to a prediction on the test set (75 directions). The latter two methods (NNLS, EBP) use the regularization parameters $\lambda = 1$ and the $c$ chosen by cross-validated NNLS. Figure 2 illustrates the first two iterations of EBP applied to these simulated data. The estimated fODF are compared to the true fODF by the antipodally symmetrized Earth Mover's

distance (EMD) [15] in each iteration. Figure 3 demonstrates the progress of the internal state of the EBP algorithm in many repetitions of the simulation. In the simulation results (Figure 4), EBP clearly reaches a more accurate solution than DTI, and a sparser solution than NNLS.

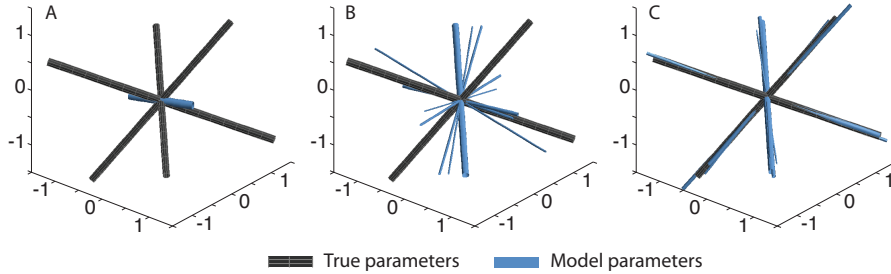

Figure 4: DWI Simulation results. Ground truth entered into the simulation is a configuration of 3 crossing fascicles (A). DTI estimates a single primary diffusion direction that coincides with none of these directions (B). NNLS estimates an fODF with many, demonstrating the discretization error (see also Figure 1). EBP estimates a much sparser solution with weights concentrated around the true peaks (D).

The same procedure is used to fit the three models to DWI data, obtained at 2x2x2 $mm^3$, at a b-value of 4000 $s/mm^2$. In the these data, the true fODF is not known. Hence, only test prediction error can be obtained. We compare RMSE of prediction error between the models in a region of interest (ROI) in the brain containing parts of the corpus callosum, a large fiber bundle that contains many fibers connecting the two hemispheres, as well as the centrum semiovale, containing multiple crossing fibers (Figure 5). NNLS and EBP both have substantially reduced error, relative to DTI.

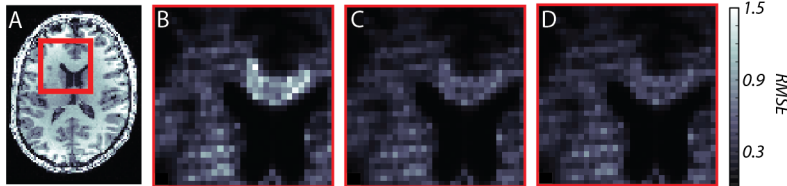

Figure 5: DWI data from a region of interest (A, indicated by red frame) is analyzed and RMSE is displayed for DTI (B), NNLS(C) and EBP(D).

## 4 Conclusions

We developed an algorithm to model multi-dimensional mixtures. This algorithm, *Elastic Basis Pursuit* (EBP), is a combination of principles from boosting, and principles from the Lawson-Hanson *active set* algorithm. It fits the data by iteratively generating and testing the match of a set of candidate kernels to the data. Kernels are added and removed from the set of candidates as needed, using a totally corrective backfitting step, based on the match of the entire set of kernels to the data at each step. We show that the algorithm reaches the global optimum, with fixed iteration guarantees. Thus, it can be practically applied to separate a multi-dimensional signal into a sum of component signals. For example, we demonstrate how this algorithm can be used to fit diffusion-weighted MRI signals into nerve fiber fascicle components.

### Acknowledgments

The authors thank Brian Wandell and Eero Simoncelli for useful discussions. CZ was supported through an NIH grant 1T32GM096982 to Robert Tibshirani and Chiara Sabatti, AR was supported through NIH fellowship F32-EY022294. FP was supported through NSF grant BCS1228397 to Brian Wandell

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
