[Supplementary Material]

# 1 Deconvolution of High Dimensional Mixtures via Boosting, with Application to Diffusion-Weighted MRI of Human Brain: Supplemental Material

## 1.1 Definitions

In this paper, we restrict our attention to the use of squared-error loss; resulting in penalized least-squares problem

$$\text{minimize}_{\hat{K},\hat{w},\hat{\boldsymbol{\theta}}} \left\| y_i - \sum_{k=1}^{\hat{K}} \hat{w}_k f_{\hat{\theta}_k}(x_i) \right\|^2 + \lambda P_{\boldsymbol{\theta}}(w) \tag{1}$$

where $P_{\boldsymbol{\theta}}(w)$ is a *convex* penalty function of $(\boldsymbol{\theta}, w)$.

Recall that one can define *convex functions* on $(w, \boldsymbol{\theta}) \in \bigcup_{K=1}^{\infty}[0,\infty)^K \times \Theta^K$ in the following manner. First, define a *sum* of $w^i = (w_i^i, \ldots, w_{K^i}^i)$, $\boldsymbol{\theta}^i = (\theta_1^i, \ldots, \theta_{K^i}^i)$ for $i = 1, \ldots, L$ by

$$\sum_{i=1}^{L}(w^i, \boldsymbol{\theta}^i) = (w, \boldsymbol{\theta}) \tag{2}$$

$$w = (w_1^1, \ldots, w_{K^1}^1, w_1^2, \ldots, w_{K^2}^2, \ldots, \ldots, w_1^L, \ldots, w_{K^L}^L) \tag{3}$$

$$\boldsymbol{\theta} = (\theta_1^1, \ldots, \theta_{K^1}^1, \theta_1^2, \ldots, \theta_{K^2}^2, \ldots, \ldots, \theta_1^L, \ldots, \theta_{K^L}^L) \tag{4}$$

and *scalar product* by

$$\alpha(w^1, \boldsymbol{\theta}^1) = (w, \boldsymbol{\theta}^1) \tag{5}$$

$$w = (\alpha w_1^1, \ldots, \alpha w_{K^1}^1) \tag{6}$$

for $\alpha \geq 0$. Then a *convex function* $G((w, \boldsymbol{\theta}))$ satisfies

$$G\left(\sum_{i=1}^{L} \alpha_i(w^i, \boldsymbol{\theta}^i)\right) \leq \sum_{i=1}^{L} \alpha_i G((w^i, \boldsymbol{\theta}^i))$$

For our convergence results to hold, we require an oracle function $\tau : \mathbb{R}^n \to \Theta$ which satisfies

$$\left\langle \tilde{r}, \frac{\tilde{f}_{\tau(\tilde{r})}}{||\tilde{f}_{\tau(\tilde{r})}||} \right\rangle \geq \alpha \rho(\tilde{r}) \tag{7}$$

where

$$\rho(\tilde{r}) = \sup_{\theta \in \Theta} \left\langle \tilde{r}, \frac{\tilde{f}_{\theta}}{||\tilde{f}_{\theta}||} \right\rangle \tag{8}$$

for some fixed $\alpha > 0$. Our algorithm will also work with a stochastic oracle that satisfies (7) with fixed probability $p > 0$ for every input $r$.

## 1.2 Regularization

*(An expanded version of the section 2.1 of the main paper.)*

A variety of $L_1$-norm based penalty functions can be accommodated by EBP, by using a modified input $\tilde{y}$ and kernel function family $\tilde{f}_{\theta}$, so that

$$\text{argmin}_{K,w,\boldsymbol{\theta}} \left\| \tilde{y} - \sum_{i=1}^{K} \tilde{f}_{\theta} \right\|^2 = \text{argmin}_{K,w,\boldsymbol{\theta}} \left\| y - \sum_{i=1}^{K} \vec{f}_{\theta} \right\|^2 + \lambda P_{\boldsymbol{\theta}}(w)$$

We will use our modified $L_2$Boost algorithm to produce a path of solutions for objective function on the left side, which results in a solution path for the penalized objective function (1).

Firstly, it is possible to embed the penalty $P_{\boldsymbol{\theta}}(w) = ||w||_1^2$ in the optimization problem (1). One can show that solutions obtained by using the penalty function $P_{\boldsymbol{\theta}}(w) = ||w||_1^2$ have a one-to-one correspondence with solutions of obtained using the usual $L_1$ penalty $||w||_1$. The penalty $||w||_1^2$ is implemented by calling EBP on modified input $\tilde{y} = \begin{pmatrix} y \\ 0 \end{pmatrix}$ and using modified kernel vectors $\tilde{f}_\theta = \begin{pmatrix} \vec{f}_\theta \\ \sqrt{\lambda} \end{pmatrix}$.

While the $L_1$ penalization imposes the same penalty for every $\theta$, a useful alternative can be useful to impose a "roughness" penalty $g(\theta)$ on the kernel functions, so that smoother kernel functions are preferred. For example, one might consider the first derivative penalty

$$g(\theta) = \int_x ||\nabla_x f_\theta(x)||^2 dx$$

or an approximation

$$g(\theta) = \frac{1}{n} \sum_{i=1}^{n} ||\nabla_x f_\theta(x_i)||^2$$

The mixture model can be fit using the penalty $P_{\boldsymbol{\theta}}(w) = \left\| \sum_{i=1}^{K} g(\theta_i) w_i \right\|^2$. This is done by setting $\tilde{y} = \begin{pmatrix} y \\ 0 \end{pmatrix}$ and $\tilde{f}_\theta = \begin{pmatrix} \vec{f}_\theta \\ \sqrt{\lambda} g(\theta) \end{pmatrix}$

For *unweighted* mixture problems, one can enforce the constraint $||w||_1 = 1$ by means of the penalty $P_{\boldsymbol{\theta}}(w) = (1 - ||w||_1)^2$. This is implemented using $\tilde{y} = \begin{pmatrix} y \\ \sqrt{\lambda} \end{pmatrix}$, $\tilde{f}_\theta = \begin{pmatrix} \vec{f}_\theta \\ \sqrt{\lambda} \end{pmatrix}$. As $\lambda \to \infty$, one obtains a hard constraint.

For all of the above penalties, the sample size in the transformed problem $\tilde{n}$, is one plus the sample size of the original problem, $n$.

Finally, *nonnegative* kernel functions satisfying $f_\theta(x) \geq 0$ satisfy a *self-regularizing* property [Slawski], so that additional penalization is optional. If no penalization is added, we take $\tilde{y} = y$ and $\tilde{f}_\theta = \vec{f}_\theta$, so $\tilde{n} = n$. For our fast convergence results, *either* nonnegativity of the kernel function *or* imposition of one of the above penalties will suffice.

In the following sections, define

$$\tilde{F}_{\boldsymbol{\theta}} = \left[ \tilde{f}_{\theta_1}, \ldots, \tilde{f}_{\theta_K} \right] \tag{9}$$

## 1.3 EBP Pseudocode

Here we present our elastic basis pursuit algorithm for producing a path of solutions $(w^{(1)}, \boldsymbol{\theta}^{(1)}), \ldots$ which progressively minimize

$$\text{minimize}_{K, w > 0, \boldsymbol{\theta}} \left\| \tilde{y} - \sum_{i=1}^{K} w_i \tilde{f}_{\theta_i} \right\|^2 \tag{10}$$

**Inputs**

- Input vector $\tilde{y} \in \mathbb{R}^{\tilde{n}}$.
- Validation function $\text{Err}_{val}(w, \boldsymbol{\theta})$ which uses a validation set to estimate the prediction error of the model $(w, \boldsymbol{\theta})$,
- Oracle $\tau : \mathbb{R}^{\tilde{n}} \to \Theta$ satisfying (7)
- Function $\tilde{f}_\theta : \Theta \to \mathbb{R}^{\tilde{n}}$ mapping parameters to regressors
- Initial estimate $(w^{(0)}, \boldsymbol{\theta}^{(0)})$ and residual $\tilde{r}^{(0)}$ obtained by using NNLS to solve (10) and then discarding any zero weights and corresponding parameters. Let $K^{(0)}$ be the number of components in $w^{(0)}$.

- Maximum number of iterations, $M$.

**Elastic Basis Pursuit**

1: **for** $m = 1, \ldots, M$ **do**
2:     $\boldsymbol{\theta}^{(m-\frac{1}{2})} \leftarrow (\tau(\tilde{r}^{(m-1)}), \theta_1^{(m-1)}, \ldots, \theta_{K^{(m-1)}}^{(m-1)})$
3:     $K^{(m-\frac{1}{2})} \leftarrow K^{(m-1)} + 1$
4:     Using NNLS, set $\beta^{(m)} \leftarrow \text{argmin}_{\beta>0} ||\tilde{y} - \tilde{F}_{\boldsymbol{\theta}^{(m-\frac{1}{2})}} \beta||^2$ and $\tilde{r}^{(m)} \leftarrow \tilde{y} - \tilde{F}_{\boldsymbol{\theta}^{(m-\frac{1}{2})}} \beta^{(m)}$
5:     $K^{(m)} \leftarrow ||\beta^{(m)}||_0$
6:     $\{i_1^{(m)}, \ldots, i_{K^{(m)}}^{(m)}\} \leftarrow \{i \in \{1, \ldots, K^{(m-\frac{1}{2})}\} : \beta_i^{(m)} \neq 0\}$
7:     $\boldsymbol{\theta}^{(m)} \leftarrow \left( \theta_{i_1^{(m)}}^{(m-1)}, \ldots, \theta_{i_{K^{(m)}}^{(m)}}^{(m-1)} \right)$
8:     $w^{(m)} \leftarrow \left( w_{i_1^{(m)}}^{(m-1)}, \ldots, w_{i_{K^{(m)}}^{(m)}}^{(m-1)} \right)$
9:     $\text{Err}_{val}^{(m)} \leftarrow \text{Err}_{val}(w^{(m)}, \boldsymbol{\theta}^{(m)})$
10: **end for**

Step 2 calls the oracle to find $\theta_{new} = \tau(\tilde{r})$, and adds $\theta_{new}$ to the active set $\boldsymbol{\theta}$. Step 4 refits the weights $w$ and updates the residual $\tilde{r}$. Step 7 prunes the active set $\boldsymbol{\theta}$ by removing any parameter $\theta$ whose weight is zero. This ensures that the active set $\boldsymbol{\theta}$ remains sparse in each iteration. Step 9 computes an estimated prediction error at each iteration, via an independent validation set. Optionally, one can add a command to stop the algorithm early when the prediction error begins to climb (indicating overfitting).

## 1.4 Proofs of Convergence

*(An expanded version of the section 2.3 of the main paper.)*

Proposition 1 establishes regularization, smoothness and compactness assumptions which ensure the existence of a maximally saturated model $(w^*, \boldsymbol{\theta}^*)$ of size $K^* \leq \tilde{n}$. Indeed, if a saturated model exists, then a saturated model with at most $\tilde{n}$ terms also exists: this is a consquence of the properties of nonnegative least squares [Lawson] . This fact is stated in Lemma 1.

**Lemma 1**. *Fix $\tilde{y} \in \mathbb{R}^{\tilde{n}}$ function $\tilde{f} \to \mathbb{R}^{\tilde{n}}$. For any positive integer $K \geq \tilde{n}$, and for any $w \in [0, \infty)^K$, $\boldsymbol{\theta} \in \Theta^K$, there exists $\tilde{w}, \tilde{\boldsymbol{\theta}} \in \Theta^{\tilde{n}}$ such that*

$$\left\| \tilde{y} - \tilde{F}_{\tilde{\boldsymbol{\theta}}} \tilde{w} \right\|^2 \leq \left\| \tilde{y} - \tilde{F}_{\boldsymbol{\theta}} w \right\|^2$$

**Proof**. By [Lawson], we can find $\beta = \text{argmin} ||\tilde{y} - \tilde{F}_{\boldsymbol{\theta}} \beta||^2$ with $||\beta||_0 \leq \tilde{n}$. Clearly,

$$\left\| \tilde{y} - \tilde{F}_{\boldsymbol{\theta}} \beta \right\|^2 \leq \left\| \tilde{y} - \tilde{F}_{\boldsymbol{\theta}} w \right\|^2$$

Let $s = ||\beta||_0$ and let

$$\{i_1, \ldots, i_s\} = \{i \in \{1, \ldots, K\} : \beta_i > 0\}$$

If $s \leq \tilde{n}$, choose $i_s, \ldots, i_{\tilde{n}}$ so that $\{i_1, \ldots, i_{\tilde{n}}\}$ has $\tilde{n}$ unique elements. Define $\tilde{w} = (\beta_{i_1}, \ldots, \beta_{i_{\tilde{n}}})$ and $\tilde{\boldsymbol{\theta}} = (\theta_{i_1}, \ldots, \theta_{i_{\tilde{n}}})$. Then

$$\left\| \tilde{y} - \tilde{F}_{\tilde{\boldsymbol{\theta}}} \tilde{w} \right\|^2 = \left\| \tilde{y} - \tilde{F}_{\boldsymbol{\theta}} \beta \right\| \leq \left\| \tilde{y} - \tilde{F}_{\boldsymbol{\theta}} w \right\|^2$$

$\square$

Having proved lemma 1, we have reduced the problem of showing the existence of a maximally saturated model to that of showing the existence of a maximally saturated model with $\tilde{n}$ components. However, we will need additional regularization assumptions.

**Proposition 1**. *Let $\tilde{y}$ be a vector in $\mathbb{R}^{\tilde{n}}$, let $\Theta$ be a compact set in $\mathbb{R}^D$, and let $\hat{f}_\theta : \Theta \to \mathbb{R}^{\tilde{n}}$ be a continuous vector-valued function with respect to $\theta$. Furthermore, assume that $\tilde{f}_\theta$ is adequately*

*regularized in the sense that there exists $\epsilon > 0$, $v \in \mathbb{R}^n$ such that*

$$\inf_{\theta \in \Theta} \langle v, \tilde{f}_\theta \rangle \geq \epsilon \tag{11}$$

*Then there exists a nonnegative integer $K^* \leq \tilde{n}$ and $w^* = (w_1^*, \ldots, w_{K^*}^*)$ and $\boldsymbol{\theta}^* = (\theta_1^*, \ldots, \theta_{K^*}^*)$, with $w^* \in [0, \infty)^{\tilde{n}}$ and $\boldsymbol{\theta}^* \in \Theta^{\tilde{n}}$ such that the residual $\tilde{r}^*$, defined by*

$$\tilde{r}^* = \tilde{y} - \tilde{F}_{\boldsymbol{\theta}^*} w^*$$

*satisfies*

$$||\tilde{r}^*||^2 = \inf_{w, \boldsymbol{\theta}, K \in \mathbb{N}} \left\| \tilde{y} - \sum_{i=1}^{K} w_i \tilde{f}_{\theta_i} \right\|^2 \tag{12}$$

The regularization condition (11) is satisfied either when $L_1$ regularization is imposed, *or* the kernels satisfy a positivity condition, i.e. $\inf_{\theta \in \Theta} f_\theta(x_i) \geq 0$ for $i = 1, \ldots, n$ and $\inf_{\theta \in \Theta} ||\vec{f}_\theta|| > 0$. Under $L_1$ regularization, $P_{\boldsymbol{\theta}}(w) = ||w||_1^2$, one can use $v = (0, 0, \ldots, 0, 1)$. Given positivity, one can use $v = (1, 1, \ldots, 1)$.

Before proving proposition 1, we will first prove a lemma stating that the regularization condition implies that any $w, \theta$ for which $||w||_1$ is large, also has a large residual.

**Lemma 2.** *Fix $\tilde{y} \in \mathbb{R}^{\tilde{n}}$ function $\tilde{f} \to \mathbb{R}^{\tilde{n}}$. Furthermore, assume that the problem is adequately regularized in the sense that there exist $\epsilon > 0$, $v \in \mathbb{R}^n$ such that (11) holds. Define*

$$U = \frac{||\tilde{y}|| \, ||v|| + \langle v, \tilde{y} \rangle}{\epsilon} \tag{13}$$

*Then for all $(w, \boldsymbol{\theta}) \in [0, \infty)^{\tilde{n}} \times \Theta^{\tilde{n}}$ with $||w||_1 > U$,*

$$\left\| \tilde{y} - \tilde{F}_{\boldsymbol{\theta}} w \right\|^2 \geq ||\tilde{y}||^2$$

**Proof.** Take $(w, \boldsymbol{\theta}) \in [0, \infty)^{\tilde{n}} \times \Theta^{\tilde{n}}$ with $||w||_1 > U$. Then

$$\langle v, \tilde{F}_{\boldsymbol{\theta}} w - \tilde{y} \rangle = -\langle v, \tilde{y} \rangle + \sum_{i=1}^{\tilde{n}} \langle v, \hat{f}_{\theta_i} \rangle w_i$$

$$\geq -\langle v, \tilde{y} \rangle + \sum_{i=1}^{\tilde{n}} \epsilon w_i$$

$$\geq -\langle v, \tilde{y} \rangle + U\epsilon$$

But by the Cauchy-Schwarz inequality

$$\left\| \tilde{y} - \tilde{F}_{\boldsymbol{\theta}} w \right\|^2 \geq \frac{\langle v, \tilde{F}_{\boldsymbol{\theta}} w - \tilde{y} \rangle^2}{||v||^2}$$

which, by our first result, is bounded below by

$$\geq \frac{(U\epsilon - \langle v, \tilde{y} \rangle)^2}{||v||^2}$$

Now applying (13),

$$\geq ||\tilde{y}||^2$$

which completes the proof. $\square$.

Having proved Lemma 2, we now know that any model which comes close to minimizing (10) must have bounded $L_1$ norm. This, in conjunction with compactness of the parameter space and continuity of $\tilde{f}_\theta$, allows us to complete the proof of proposition 1, which establishes the existence of a model which minimizes (10).

**Proof of proposition 1**. From Lemma 1, there exists a sequence of models in $(0, \infty]^{\tilde{n}} \times \Theta^{\tilde{n}}$, $(w^{[1]}, \boldsymbol{\theta}^{[1]}), \dots$ so that

$$\lim_{m \to \infty} ||\tilde{r}^{[m]}||^2 = \inf_{w, \theta, K \in \mathbb{N}} \left\| \tilde{y} - \sum_{i=1}^{K} w_i \tilde{f}_{\theta_i} \right\|^2$$

where

$$\tilde{r}^{[m]} = \tilde{y} - \tilde{F}_{\boldsymbol{\theta}^{[m]}} w^{[m]}$$

Let $U$ be as defined in Lemma 2, and choose $j \in \mathbb{R}$ so that for all $m \geq j$, $||\tilde{r}^{[m]}||^2 < ||\tilde{y}||^2$. Then by Lemma 2, for all $m > j$, $w^{[m]} \in [0, U]^{\tilde{n}}$. Since $\Theta$ is compact, so is $[0, U]^{\tilde{n}} \times \Theta^{\tilde{n}}$. Hence $\{(w^{[m]}, \boldsymbol{\theta}^{[m]})\}_{m=j}^{\infty}$ has a convergent subsequence with limiting point $w^{\infty}, \boldsymbol{\theta}^{\infty}$. By the continuity of $\tilde{f}_{\theta}$,

$$\left\| \tilde{y} - \tilde{F}_{\boldsymbol{\theta}^{\infty}} w^{\infty} \right\|^2 = \inf_{w, \theta, K \in \mathbb{N}} \left\| \tilde{y} - \sum_{i=1}^{K} w_i \tilde{f}_{\theta_i} \right\|^2$$

Taking $K^* = ||w^{\infty}||_0$, and $\{i_1, \dots, i_{K^*}\} = \{i \in \{1, \dots, \tilde{n}\} : w_i^{\infty} > 0\}$, define $w^* = (w_{i_1}^{\infty}, \dots, w_{i_{K^*}}^{\infty})$ and $\boldsymbol{\theta}^* = (\theta_{i_1}^{\infty}, \dots, \theta_{i_{K^*}}^{\infty})$. Then

$$||\tilde{r}^*||^2 = \left\| \tilde{y} - \tilde{F}_{\boldsymbol{\theta}^*} w^* \right\|^2 = \left\| \tilde{y} - \tilde{F}_{\boldsymbol{\theta}^{\infty}} w^{\infty} \right\|^2 = \inf_{w, \theta, K \in \mathbb{N}} \left\| y - \sum_{i=1}^{K} w_i \tilde{f}_{\theta_i} \right\|^2$$

as desired. □

The existence of such a saturated model $(w^*, \boldsymbol{\theta}^*)$, in conjunction with existence of the oracle $\tau$, enables us to state fixed-iteration guarantees on the precision of EBP, which implies asymptotic convergence to the global optimum.

To do so, recall the definition of the maximum correlation function $\rho$ (8), and define the quantity $\rho^{(m)} = \rho(r^{(m)})$. Proposition 2 uses the fact that the residuals $\tilde{r}^{(m)}$ are orthogonal to $\tilde{F}^{(m)}$, thanks to the NNLS fitting procedure in step 2. This allows us to bound the objective function gap in terms of $\rho^{(m)}$. Proposition 3 uses properties of the oracle $\tau$ to lower bound the progress per iteration in terms of $\rho^{(m)}$.

**Proposition 2** *Assume the conditions of Proposition 1. Take $w^*, \boldsymbol{\theta}^*$ satisfying* (12). *Then defining*

$$B^* = 2 \sum_{i=1}^{K^*} w_i^* ||\tilde{f}_{\theta_i^*}|| \tag{14}$$

*the $m$th residual of the EBP algorithm $\tilde{r}^{(m)}$ can be bounded in size by*

$$||\tilde{r}^{(m)}||^2 \leq ||\tilde{r}^*||^2 + B^* \rho^{(m)}$$

**Proof.** Define $h^{(m)} : \mathbb{R}^{K^{(m)}} \times \mathbb{R}^{K^*} \to \mathbb{R}$ by

$$h^{(m)}(a, b) = \left\| \tilde{r}^{(m)} - \sum_{i=1}^{K^{(m)}} a_i \tilde{f}_{\theta_i^{(m)}} - \sum_{i=1}^{K^*} b_i \tilde{f}_{\theta_i^*} \right\|^2$$

Since $h$ is a squared norm of a affine transformation of $(a, b)$, $h$ is convex in $(a, b)$. Also check that $h^{(m)}(0, 0) = ||\tilde{r}^{(m)}||^2$ and $h^{(m)}(-w^{(m)}, w^*) = ||\tilde{r}^*||^2$.

Since $\tilde{r}^{(m)}$ is the least squares residual of regressing $\tilde{y}$ on $\tilde{F}^{(m)}$, we have

$$\langle \tilde{r}, \tilde{f}_{\theta_i^{(m)}} \rangle = 0$$

for $i = 1, \dots, K^{(m)}$.

Therefore,

$$\frac{\partial h^{(m)}}{a_i}(0, 0) = -2 \langle \tilde{r}, \tilde{f}_{\theta_i^{(m)}} \rangle = 0$$

Meanwhile,

$$\frac{\partial h^{(m)}}{b_i}(0,0) = -2\langle \tilde{r}^{(m)}, \tilde{f}_{\theta_i^*}\rangle \geq -2\rho^{(m)}||\tilde{f}_{\theta_i^*}||$$

for $i = 1, \ldots, K^*$ by definition of $\rho^{(m)}$. Now due to the convexity of $h$, we have

$$||\tilde{r}^*||^2 = h(-w^{(m)}, w^*) \tag{15}$$

$$\geq h(0,0) - w^{(m)}\nabla_a h(0,0) + w^*\nabla_b h(0,0) \tag{16}$$

$$= h(0,0) + w^*\nabla_b h(0,0) \tag{17}$$

$$= h(0,0) + \sum_{i=1}^{K^*} w_i^* \frac{\partial h}{b_i}(0,0) \tag{18}$$

$$\geq h(0,0) + \sum_{i=1}^{K^*} w_i^*(-2\rho^{(m)}||\tilde{f}_{\theta_i^*}||) \tag{19}$$

$$= h(0,0) + -2\rho^{(m)} \sum_{i=1}^{K^*} w_i^* ||\tilde{f}_{\theta_i^*}|| \tag{20}$$

$$= ||\tilde{r}^{(m)}||^2 - B^*\rho^{(m)} \tag{21}$$

as desired. $\square$.

Proposition 3 is mainly a consequence of the fact that in a linearly constrained regression problem, adding a new variable to the regression is at least as good as fitting that variable by itself to the residual.

**Proposition 3** *Assume the conditions of Proposition 1. Then*

$$||\tilde{r}^{(m)}||^2 - ||\tilde{r}^{(m+1)}||^2 \geq (\alpha\rho^{(m)})^2$$

*which also implies that the sequence $||\tilde{r}^{(0)}||^2, \ldots$ is decreasing.*

**Proof**.

We have

$$||\tilde{r}^{(m+1)}||^2 = \min_{\beta > 0} ||\tilde{y} - \tilde{F}_{\boldsymbol{\theta}^{(m+\frac{1}{2})}}\beta||^2 \tag{22}$$

$$\leq \left\| \tilde{y} - \tilde{F}_{\boldsymbol{\theta}^{(m)}}w^{(m)} - \tilde{f}_{\theta_1^{(m+\frac{1}{2})}} \frac{\langle \tilde{f}_{\theta_1^{(m+\frac{1}{2})}}, \tilde{r}^{(m)}\rangle}{||\tilde{f}_{\theta_1^{(m+\frac{1}{2})}}||^2} \right\|^2 \tag{23}$$

$$= \left\| \tilde{r}^{(m)} - \tilde{f}_{\theta_1^{(m+\frac{1}{2})}} \frac{\langle \tilde{f}_{\theta_1^{(m+\frac{1}{2})}}, \tilde{r}^{(m)}\rangle}{||\tilde{f}_{\theta_1^{(m+\frac{1}{2})}}||^2} \right\|^2 \tag{24}$$

$$= ||\tilde{r}^{(m)}||^2 - \left\| \frac{\tilde{f}_{\theta_1^{(m+\frac{1}{2})}}}{||\tilde{f}_{\theta_1^{(m+\frac{1}{2})}}||} \frac{\langle \tilde{f}_{\theta_1^{(m+\frac{1}{2})}}, \tilde{r}^{(m)}\rangle}{||\tilde{f}_{\theta_1^{(m+\frac{1}{2})}}||} \right\|^2 \tag{25}$$

$$= ||\tilde{r}^{(m)}||^2 - \left( \frac{\langle \tilde{f}_{\theta_1^{(m+\frac{1}{2})}}, \tilde{r}^{(m)}\rangle}{||\tilde{f}_{\theta_1^{(m+\frac{1}{2})}}||} \right)^2 \tag{26}$$

$$\leq ||\tilde{r}^{(m)}||^2 - (\alpha\rho^{(m)})^2 \tag{27}$$

Here, (23) follows from the fact that $\tilde{F}_{\boldsymbol{\theta}^{(m+\frac{1}{2})}} = \left[ \tilde{f}_{\theta_1^{(m+\frac{1}{2})}} \quad \tilde{F}_{\boldsymbol{\theta}^{(m)}} \right]$ and (24) follows from the fact that $\tilde{r}^{(m)} = \tilde{y} - \tilde{F}_{\boldsymbol{\theta}^{(m)}} w^{(m)}$. Next, (25) is obtained by an application of the Pythagorean theorem, and (27) by applying the definitions of $\rho^{(m)}$ and the condition (7) on $\tau$. $\square$

Combining Propositions 2 and 3 yields our main result for the non-asymptotic convergence rate.

**Proposition 4** *Assume the conditions of Proposition 1. Then for all $m > 0$,*

$$||\tilde{r}^{(m)}||^2 - ||\tilde{r}^*||^2 \leq \frac{B_{min}\sqrt{||\tilde{r}^{(0)}||^2 - ||\tilde{r}^*||^2||}}{\alpha} \frac{1}{\sqrt{m}}$$

*where*

$$B_{min} = \inf_{w^*, \boldsymbol{\theta}^*} B^*$$

*for $B^*$ defined in* (14)

**Proof.** Take $(w^*, \boldsymbol{\theta}^*)$ satisfying (12), and define $B^*$ as in (14). Define $g_i = ||\tilde{r}^{(i)}||^2 - ||\tilde{r}^*||^2$ for $i = 0, \dots$ and fix $m \in \mathbb{N}$. By Proposition 2,

$$g_m = ||\tilde{r}^{(m)}||^2 - ||\tilde{r}^*||^2 \leq B^* \rho^{(m)}$$

By Proposition 3, $g_0 \geq g_1 \geq \cdots$, so that for all $0 \leq i \leq m$,

$$\rho_{(i)} \geq \frac{g_i}{B^*} \geq \frac{g_m}{B^*} \tag{28}$$

Now observe that

$$\begin{aligned} g_0 &= ||\tilde{r}^{(0)}||^2 - ||\tilde{r}^*||^2 \\ &= ||\tilde{r}^{(0)}||^2 - ||\tilde{r}^{(m)}||^2 + ||\tilde{r}^{(m)}||^2 - ||\tilde{r}^*||^2 \\ &= ||\tilde{r}^{(0)}||^2 - ||\tilde{r}^{(m)}||^2 + g_m \\ &= g_m + \sum_{i=1}^{m-1} ||\tilde{r}^{(i)}||^2 - ||\tilde{r}^{(i+1)}||^2 \end{aligned}$$

which by Proposition 3

$$\geq g_m + \sum_{i=1}^{m-1} (\alpha \rho^{(i)})^2$$

Applying (28),

$$\geq g_m + \sum_{i=1}^{m-1} \left( \frac{\alpha}{B^*} g_m \right)^2$$

$$= g_m + m \left( \frac{\alpha}{B^*} \right)^2 g_m$$

Defining $C = (\alpha/B^*)^2$

$$= g_m + C m g_m^2$$

Hence

$$g_m^2 + \frac{g_m}{Cm} \leq \frac{g_0}{Cm}$$

$$g_m^2 + \frac{g_m}{Cm} + \frac{1}{(2Cm)^2} \leq \frac{g_0}{Cm} + \frac{1}{(2Cm)^2}$$

$$\left( g_m + \frac{1}{2Cm} \right)^2 \leq \frac{g_0}{Cm} + \frac{1}{(2Cm)^2} \leq \left( \sqrt{\frac{g_0}{Cm}} + \frac{1}{2Cm} \right)^2$$

$$g_m + \frac{1}{2Cm} \leq \sqrt{\frac{g_0}{Cm}} + \frac{1}{2Cm}$$

$$g_m \leq \sqrt{\frac{g_0}{Cm}} = \sqrt{\frac{g_0 (B^*)^2}{\alpha^2 m}} = \frac{B^*}{\alpha} \frac{\sqrt{g_0}}{\sqrt{m}}$$

The proof follows by noting that $g_m \leq \frac{B^*}{\alpha} \frac{\sqrt{g_0}}{\sqrt{m}}$ holds for any choice of $(w^*, \boldsymbol{\theta}^*)$. $\square$