[Reviews · NeurIPS 2014]

Submitted by Assigned_Reviewer_5

This paper proposes elastic basis pursuit, which is an algorithm for fitting sparse mixture models via convex optimisation. The method combines boosting with an active set method for continuous basis pursuit. The authors provide proofs of convergence of the algorithm and test it experimentally within the application of estimating fibre directions from DWI data.

The algorithm is well motivated, and the authors clearly understand well the wider space of solutions to this family of problem and their pros and cons. It is also well constructed and combines the benefits of different existing approaches retaining computational efficiency, convexity, and accuracy of solution. I am not intimately familiar with the latest literature on this problem, but as far as I am aware, the algorithm is novel and represents a fairly minor but not insignificant advance over the current state of the art. I have not checked the details of the proofs of convergence, but, assuming they are correct, their presence adds to the value of the paper.

The chosen application to demonstrate the technique is timely and appropriate, as diffusion MRI is used widely for estimating microstructural features of brain tissue, including fibre orientations, which then leads to algorithms that reconstruct global brain connectivity and are widely used in neuroscience and neurology. Resolving the orientations of crossing fibres is a long standing problem, which saw a lot of attention from 2000-2010, and still has outstanding issues so improved solutions have significant value. However, the experimental results presented here are very limited. They show one simulated example, for which the proposed algorithm will clearly perform better than the others tested, as well as a largely qualitative result comparing fitting error maps over a small brain region. The simulation results would be much stronger if they included a quantitative evaluation of how much better and how often the new algorithm does in comparison to NNLS and other algorithms used commonly in the diffusion MRI literature. The brain-data results show no obvious difference between the NNLS and EBP output, although presumably there are significant differences. Maps of number of fibre directions or subsequent tractography would add a great deal to convince the reader that the new fitting algorithm really makes a difference worth thinking about. Even a simple test of the significance of the difference in error scores between NNLS and EBP would add something.

Minor points:

page 6. The diffusion tensor in 3D is rank 3 in general.

page 7. The form of the oracle function is not clear.

page 7. Should w1, w2, w3 sum to 1? Or does the value of c ensure the sum of compartment fractions sums to 1? Either way, where is the constraint imposed? Also, it is unclear how c represents an isotropic diffusion component.

page 8. Figures 4 and 5 not referred to from text; fig 4 erroneously referred to as fig 3, but fig 5 never mentioned.

Summary: The basic mixture-model fitting method seems reasonable and solidly analysed theoretically. The experimental work is very preliminary and not compelling.

Submitted by Assigned_Reviewer_26

This paper proposes an algorithm called elastic basis pursuit for fitting multi-dimensional mixture models and demonstrates advantages of the proposed method over standard approaches, such as diffusion tensor imaging, or DTI (not a mixture model) and non-negative least-squares (NNLS) method of fitting mixtures, on diffusion-weighted MRI data. Simulations show clear improvement over both baseline approaches, and real-life diffusion MRI data also suggest that proposed method, as well as NNLS, yield considerably lower errors than the standard DTI. The paper proves the convergence of the proposed algorithm, which essentilly is an augmented boosting approach, where the active set does not necessarily grow monotonically, i.e. active components might be deleted in future iterations, since the model weights of ALL components are refitted at each iteration. It would be nice to elucidate relationship of the proposed approach to orthogonal matching pursuit and forward-selection in regression, which both have the above refitting step, as opposed to simpler matching pursuit or forward stagewise regression.
Summary: This is a nicely written paper, the proposed algorithm demonstrates improvement over state-of-art DTI on both simulated and real data, and theoretical results on the convergence appear to be sound. It looks like a useful addition to the state-of-art on diffusion-weighted MRI.

Submitted by Assigned_Reviewer_40

The authors propose an algorithm to fit multi-dimensional mixtures, called elastic basis pursuit (EBP). It uses the principles of L2-boost, together with refitting of the weights and pruning of the parameters. Kernels are added and removed from the set of candidates based on the match of the entire set of kernels to the data at each step, based on a totally corrective backfitting step. The algorithm can be applied to separate a multi-dimensional signal into a sum of component signals. For example and as demonstrated, it can be used to fit diffusion-weighted MRI signals into nerve fiber fascicle components.

Can the authors comment on the limitations and computation cost of the proposed algorithm? Also, further validation would make the method more convincing.
Summary: This paper presents an algorithm for modeling multi-dimensional mixtures, based on the principles of L2-boost together with refitting of the weights and pruning of the parameters. The authors demonstrate the utility of the method in DWI data.
Author Feedback
Author rebuttal: We thank the reviewers for their constructive criticism and supportive comments. First, we will address the common concern of the reviewers: that we show how much and how often EBP does better than other algorithms (NNLS, and the state of the art in the field) in simulations and in dMRI data. First, we emphasize that the NNLS procedure used here differs from previous approaches to fitting dMRI and is our novel, previously unpublished procedure. This approach is more accurate than the state-of-the-art algorithms in the field (e.g. CSD; Tournier et al. 2007) in fitting dMRI data (unpublished observations), and EBP does equally well. Thus, both EBP and NNLS do better than the state-of-the-art in dMRI with respect to prediction error. Importantly, in the simulations, we have found that EBP also predicts the true angle more accurately than our NNLS procedure does. This was quantified using a variant of the Earth-mover’s distance (EMD) comparing the estimated fiber orientation distribution function (fODF) with the fODF entered into the simulation included in the supplementary materials (mean EMD(EBP) = 0.1501 +/- 0.004, mean EMD(NNLS)=0.1854+/-0.002). Note that since computation of the EMD requires a ground truth, we cannot quantify EMD error in the real data.

Additional concerns:

Assigned_Reviewer_26 asked about the relationship of the proposed approach to orthogonal matching pursuit and forward-selection in regression, as opposed to simpler matching pursuit or forward stagewise regression. We emphasize that the fundamental novelty of our approach is that it uses a continuously parameterized basis rather than a finite dictionary; all previous approaches including orthogonal matching pursuit, forward-selection, matching pursuit and forward stagewise regression use a finite basis. Thus, our method is an extension of orthogonal matching pursuit to an infinite dictionary.

Assigned_Reviewer_40 asked about limitations of our approach and computation cost. The main limitation of our approach is that it does not allow the user to specify the sparsity of the final solution in advance: this is in contrast to best-K subset regression, which has a sparsity parameter K which can be fixed in advance. If the true number of components in the model is known in advance, it may be advantageous to use best-K subset regression rather than our method. However, even in that case, due to the computational cost of best-K regression, it may still be beneficial to use EBP to pre-select directions to use for best-K subset regression.

As demonstrated in our theoretical results and as seen in practice, the computational cost of our method is linear in n, the number of observations, and quadratic in the numerical precision desired. That is, to decrease numerical error by a factor of 2, the computational cost increases by a factor of 4.

Assigned_Reviewer_5 asked about maps of number of fibre directions or subsequent tractography. We are currently working on these issues, but did not yet have results to show at the time of paper submission.

Additional questions:

“page 6. The diffusion tensor in 3D is rank 3 in general.”

Response: The column rank of the matrix can indeed be (as the matrix is a 3x3 matrix), but the tensor rank is determined by the number of indices (for example, see: http://mathworld.wolfram.com/TensorRank.html). For that reason, we refer to the DTI tensor as a rank 2 tensor.

“page 7. The form of the oracle function is not clear.”

Response: Our apologies – this should have been clearer in the original submission: we used Newton-Raphson with random restarts.

“page 7. Should w1, w2, w3 sum to 1? Or does the value of c ensure the sum of compartment fractions sums to 1? Either way, where is the constraint imposed? Also, it is unclear how c represents an isotropic diffusion component. “

Response: The w’s do not sum to 1. Instead, we use regularization to make sure that they sum to approximately equal to c. This quantity is equal to the maximal diffusion-weighted signal generated by a single fiber. We refer to c as an isotropic component, because for a given b value, and diffusivity parameters defining the response function used, this quantity is then independent of number and directions of fibers in the specific voxel.

“page 8. Figures 4 and 5 not referred to from text; fig 4 erroneously referred to as fig 3, but fig 5 never mentioned.”

Response: That it correct – thanks for noticing – we will correct these typographical errors.